# Training and Analyzing Deep Recurrent Neural Networks

**Michiel Hermans, Benjamin Schrauwen**
Ghent University, ELIS departement
Sint Pietersnieuwstraat 41,
9000 Ghent, Belgium
`michiel.hermans@ugent.be`

## Abstract

Time series often have a temporal hierarchy, with information that is spread out over multiple time scales. Common recurrent neural networks, however, do not explicitly accommodate such a hierarchy, and most research on them has been focusing on training algorithms rather than on their basic architecture. In this paper we study the effect of a hierarchy of recurrent neural networks on processing time series. Here, each layer is a recurrent network which receives the hidden state of the previous layer as input. This architecture allows us to perform hierarchical processing on difficult temporal tasks, and more naturally capture the structure of time series. We show that they reach state-of-the-art performance for recurrent networks in character-level language modeling when trained with simple stochastic gradient descent. We also offer an analysis of the different emergent time scales.

## 1 Introduction

The last decade, machine learning has seen the rise of neural networks composed of multiple layers, which are often termed *deep neural networks* (DNN). In a multitude of forms, DNNs have shown to be powerful models for tasks such as speech recognition [17] and handwritten digit recognition [4]. Their success is commonly attributed to the *hierarchy* that is introduced due to the several layers. Each layer processes some part of the task we wish to solve, and passes it on to the next. In this sense, the DNN can be seen as a processing pipeline, in which each layer solves a part of the task before passing it on to the next, until finally the last layer provides the output.

One type of network that debatably falls into the category of deep networks is the recurrent neural network (RNN). When folded out in time, it can be considered as a DNN with indefinitely many layers. The comparison to common deep networks falls short, however, when we consider the *functionality* of the network architecture. For RNNs, the primary function of the layers is to introduce memory, *not* hierarchical processing. New information is added in every 'layer' (every network iteration), and the network can pass this information on for an indefinite number of network updates, essentially providing the RNN with unlimited memory depth. Whereas in DNNs input is only presented at the bottom layer, and output is only produced at the highest layer, RNNs generally receive input and produce output at each time step. As such, the network updates do not provide hierarchical processing of the information *per se*, only in the respect that older data (provided several time steps ago) passes through the recursion more often. There is no compelling reason why older data would require more processing steps (network iterations) than newly received data. More likely, the recurrent weights in an RNN learn during the training phase to *select* what information they need to pass onwards, and what they need to discard. Indeed, this quality forms the core motivation of the so-called *Long Short-term memory* (LSTM) architecture [11], a special form of RNN.

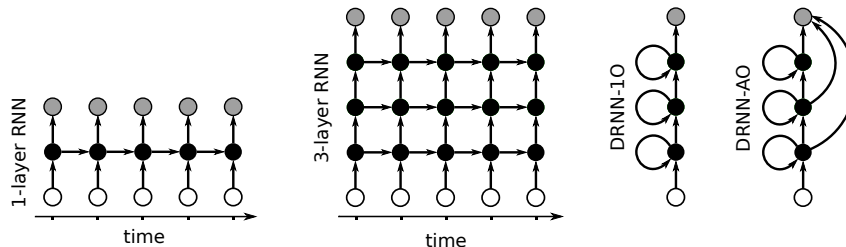

Figure 1: Schematic illustration of a DRNN. Arrows represent connection matrices, and white, black and grey circles represent input frames, hidden states, and output frames respectively. **Left:** Standard RNN, folded out in time. **Middle:** DRNN of 3 layers folded out in time. Each layer can be interpreted as an RNN that receives the time series of the previous layer as input. **Right:** The two alternative architectures that we study in this paper, where the looped arrows represent the recurrent weights. Either only the top layer connects to the output (DRNN-1O), or all layers do (DRNN-AO).

One potential weakness of a common RNN is that we may need complex, hierarchical processing of the *current* network input, but this information only passes through one layer of processing before going to the output. Secondly, we may need to process the time series at several time scales. If we consider for example speech, at the lowest level it is built up of phonemes, which exist on a very short time-scale. Next, on increasingly longer time scales, there are syllables, words, phrases, clauses, sentences, and at the highest level for instance a full conversation. Common RNNs do not explicitly support multiple time scales, and any temporal hierarchy that is present in the input signal needs to be embedded implicitly in the network dynamics.

In past research, some hierarchical architectures employing RNNs have been proposed [3, 5, 6]. Especially [5] is interesting in the sense that they construct a hierarchy of RNNs, which all operate on different time-scales (using subsampling). The authors limit themselves to artificial tasks, however. The architecture we study in this paper has been used in [8]. Here, the authors employ stacked bi-directional LSTM networks, and train it on the TIMIT phoneme dataset [7] in which they obtain state-of-the-art performance. Their paper is strongly focused on reaching good performance, however, and little analysis on the actual contribution of the network architecture is provided.

The architecture we study in this paper is essentially a common DNN (a multilayer perceptron) with temporal feedback loops in each layer, which we call a *deep recurrent neural network* (DRNN). Each network update, new information travels up the hierarchy, and temporal context is added in each layer (see Figure 1). This basically combines the concept of DNNs with RNNs. Each layer in the hierarchy is a recurrent neural network, and each subsequent layer receives the hidden state of the previous layer as input time series. As we will show, stacking RNNs automatically creates different time scales at different levels, and therefore a temporal hierarchy.

In this paper we will study character-based language modelling and provide a more in-depth analysis of how the network architecture relates to the nature of the task. We suspect that DRNNs are well-suited to capture temporal hierarchies, and character-based language modeling is an excellent real-world task to validate this claim, as the distribution of characters is highly nonlinear and covers both short- and long-term dependencies. As we will show, DRNNs embed these different timescales directly in their structure, and they are able to model long-term dependencies. Using only stochastic gradient descent (SGD) we are able to get state-of-the-art performance for recurrent networks on a Wikipedia-based text corpus, which was previously only obtained using the far more advanced Hessian-free training algorithm [19].

## 2 Deep RNNs

### 2.1 Hidden state evolution

We define a DRNN with $L$ layers, and $N$ neurons per layer. Suppose we have an input time series $\mathbf{s}(t)$ of dimensionality $N_{in}$, and a target time series $\mathbf{y}^*(t)$. In order to simplify notation we will not explicitly write out bias terms, but augment the corresponding variables with an element equal to

one. We use the notation $\bar{\mathbf{x}} = [\mathbf{x}; 1]$.

We denote the hidden state of the $i$-th layer with $\mathbf{a}_i(t)$. Its update equation is given by:

$$\mathbf{a}_i(t) = \tanh\left(\mathbf{W}_i\mathbf{a}_i(t-1) + \mathbf{Z}_i\bar{\mathbf{a}}_{i-1}(t)\right) \quad \text{if } i > 1$$
$$\mathbf{a}_i(t) = \tanh\left(\mathbf{W}_i\mathbf{a}_i(t-1) + \mathbf{Z}_i\bar{\mathbf{s}}(t)\right) \quad \text{if } i = 1.$$

Here, $\mathbf{W}_i$ and $\mathbf{Z}_i$ are the recurrent connections and the connections from the lower layer or input time series, respectively. A schematic drawing of the DRNN is presented in Figure 1.

Note that the network structure inherently offers different time scales. The bottom layer has fading memory of the input signal. The next layer has fading memory of the hidden state of the bottom layer, and consequently a fading memory of the input which reaches further in the past, and so on for each additional layer.

## 2.2 Generating output

The task we consider in this paper is a classification task, and we use a softmax function to generate output. The DRNN generates an output which we denote by $\mathbf{y}(t)$. We will consider two scenarios: that where only the highest layer in the hierarchy couples to the output (DRNN-1O), and that where all layers do (DRNN-AO). In the two respective cases, $\mathbf{y}(t)$ is given by:

$$\mathbf{y}(t) = \text{softmax}\left(\mathbf{U}\bar{\mathbf{a}}_L(t)\right), \tag{1}$$

where $\mathbf{U}$ is the matrix with the output weights, and

$$\mathbf{y}(t) = \text{softmax}\left(\sum_{i=1}^{L}\mathbf{U}_i\bar{\mathbf{a}}_i(t)\right), \tag{2}$$

such that $\mathbf{U}_i$ corresponds to the output weights of the $i$-th layer. The two resulting architectures are depicted in the right part of Figure 1.

The reason that we use output connections at each layer is twofold. First, like any deep architecture, DRNNs suffer from a pathological curvature in the cost function. If we use backpropagation through time, the error will propagate from the top layer down the hierarchy, but it will have diminished in magnitude once it reaches the lower layers, such that they are not trained effectively. Adding output connections at each layer amends this problem to some degree as the training error reaches all layers directly.

Secondly, having output connections at each layer provides us with a crude measure of its role in solving the task. We can for instance measure the decay of performance by leaving out an individual layer's contribution, or study which layer contributes most to predicting characters in specific instances.

## 2.3 Training setup

In all experiments we used stochastic gradient descent. To avoid extremely large gradients near bifurcations, we applied the often-used trick of normalizing the gradient before using it for weight updates. This simple heuristic seems to be effective to prevent gradient explosions and sudden jumps of the parameters, while not diminishing the end performance. We write the number of batches we train on as $T$. The learning rate is set at an initial value $\eta_0$, and drops linearly with each subsequent weight update. Suppose $\boldsymbol{\theta}(j)$ is the set of all trainable parameters after $j$ updates, and $\nabla_{\boldsymbol{\theta}}(j)$ is the gradient of a cost function w.r.t. this parameter set, as computed on a randomly sampled part of the training set. Parameter updates are given by:

$$\boldsymbol{\theta}(j+1) = \boldsymbol{\theta}(j) - \eta_0\left(1 - \frac{j}{T}\right)\frac{\nabla_{\boldsymbol{\theta}}(j)}{||\nabla_{\boldsymbol{\theta}}(j)||}. \tag{3}$$

In the case where we use output connections at the top layer only, we use an incremental layer-wise method to train the network, which was necessary to reach good performance. We add layers one by one and at all times an output layer only exists at the current top layer. When adding a layer, the previous output weights are discarded and new output weights are initialised connecting from the new top layer. In this way each layer has at least some time during training in which it is directly

coupled to the output, and as such can be trained effectively. Over the course of each of these training stages we used the same training strategy as described before: training the full network with BPTT and linearly reducing the learning rate to zero before a new layer is added. Notice the difference to common layer-wise training schemes where only a single layer is trained at a time. We always train the full network after each layer is added.

## 3   Text prediction

In this paper we consider next character prediction on a Wikipedia text-corpus [19] which was made publicly available[1]. The total set is about 1.4 billion characters long, of which the final 10 million is used for testing. Each character is represented by one-out-of-N coding. We used 95 of the most common characters[2] (including small letters, capitals, numbers and punctuation), and one 'unknown' character, used to map any character not part of the 95 common ones, e.g. Cyrillic and Chinese characters. We need time in the order of 10 days to train a single network, largely due to the difficulty of exploiting massively parallel computing for SGD. Therefore we only tested three network instantiations[3]. Each experiment was run on a single GPU (NVIDIA GeForce GTX 680, 4GB RAM).

The task is as follows: given a sequence of text, predict the probability distribution of the next character. The used performance metric is the average number of bits-per-character (BPC), given by $\mathrm{BPC} = -\langle \log_2 p_c \rangle$, where $p_c$ is the probability as predicted by the network of the correct next character.

### 3.1   Network setups

The challenge in character-level language modelling lies in the great diversity and sheer number of words that are used. In the case of Wikipedia this difficulty is exacerbated due to the large number of names of persons and places, scientific jargon, etc. In order to capture this diversity we need large models with many trainable parameters.

All our networks have a number of neurons selected such that in total they each had approximately 4.9 million trainable parameters, which allowed us to make a comparison to other published work [19]. We considered three networks: a common RNN (2119 units), a 5-layer DRNN-1O (727 units per layer), and a 5-layer DRNN-AO (706 units per layer)[4]. Initial learning rates $\eta_0$ were chosen at 0.5, except for the the top layer of the DRNN-1O, where we picked $\eta_0 = 0.25$ (as we observed that the nodes started to saturate if we used a too high learning rate).

The RNN and the DRNN-AO were trained over $T = 5 \times 10^5$ parameter updates. The network with output connections only at the top layer had a different number of parameter updates per training stage, $T = \{0.5, 1, 1.5, 2, 2.5\} \times 10^5$, for the 5 layers respectively. As such, for each additional layer the network is trained for more iterations. All gradients are computed using backpropagation through time (BPTT) on 75 randomly sampled sequences in parallel, drawn from the training set. All sequences were 250 characters long, and the first 50 characters were disregarded during the backwards pass, as they may have insufficient temporal context. In the end the DRNN-AO sees the full training set about 7 times in total, and the DRNN-1O about 10 times.

The matrices $\mathbf{W}_i$ and $\mathbf{Z}_{i>1}$ were initialised with elements drawn from $\mathcal{N}(0, N^{-1/2})$. The input weights $\mathbf{Z}_1$ were drawn from $\mathcal{N}(0, 1)$. We chose to have the same number of neurons for every layer, mostly to reduce the number of parameters that need to be optimised. Output weights were always initialised on zero.

| Model | BPC test |
|---|---|
| RNN | 1.610 |
| DRNN-AO | 1.557 |
| DRNN-1O | 1.541 |
| MRNN | 1.55 |
| PAQ | 1.51 |
| Hutter Prize (current record) [12] | 1.276 |
| Human level (estimated) [18] | 0.6 – 1.3 |

Table 1: Results on the Wikipedia character prediction task. The first three numbers are our measurements, the next two the results on the same dataset found in [19]. The bottom two numbers were not measured on the same text corpus.

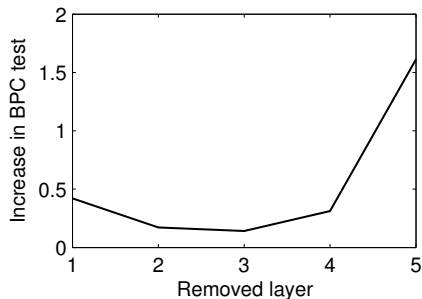

Figure 2: Increase in BPC on the test set from removing the output contribution of a single layer of the DRNN-AO.

## 3.2 Results

**Performance and text generation**

The resulting BPCs for our models and comparative results in literature are shown in Table 1. The common RNN performs worst, and the DRNN-1O the best, with the DRNN-AO slightly worse. Both DRNNs perform well and are roughly similar to the state-of-the-art for recurrent networks with the same number of trainable parameters[5], which was established with a multiplicative RNN (MRNN), trained with Hessian-free optimization in the course of 5 days on a cluster of 8 GPUs[6]. The same authors also used the PAQ compression algorithm [14] as a comparison, which we included in the list. In the table we also included two results which were not measured on the same dataset (or even using the same criteria), but which give an estimation of the true number of BPC for natural text.

To check how each layer influences performance in the case of the DRNN-AO, we performed tests in which the output of a single layer is set to zero. This can serve as a sanity check to ensure that the model is efficiently trained. If for instance removing the top layer output contribution does not significantly harm performance, this essentially means that it is redundant (as it does no preprocessing for higher layers). Furthermore we can use this test to get an overall indication of which role a particular layer has in producing output. Note that these experiments only have a limited interpretability, as the individual layer contributions are likely not independent. Perhaps some layers provide strong negative output bias which compensates for strong positive bias of another, or strong synergies might exists between them.

First we measure the increase in test BPC by removing a single layer's output contribution, which can then be used as an indicator for the importance of this layer for directly generating output. In Figure 2 we show the result. The contribution of the top layer is the most important, and that of the bottom layer second important. The intermediate layers contribute less to the direct output and seem to be more important in preprocessing the data for the top layer.

As in [19], we also used the networks in a generative mode, where we use the output probabilities of the DRNN-AO to recursively sample a new input character in order to complete a given sentence. We too used the phrase "The meaning of life is ". We performed three tests: first we generated text with an intact network, next we see how the text quality deteriorates when we leave out the contributions of the bottom and top layer respectively[7] (by setting it equal to zero before adding up

The meaning of life is the "decorator of Rose". The Ju along with its perspective character survive, which coincides with his eromine, water and colorful art called "Charles VIII".??In "Inferno" (also 220: "The second Note Game Magazine", a comic at the Old Boys at the Earl of Jerusalem for two years) focused on expanded domestic differences from 60 mm Oregon launching, and are permitted to exchange guidance.

The meaning of life is impossible to unprecede ?Pok.{* PRER)!—KGOREMFHEAZ‗CTX=R‗M —S=6‗5?&+——=7xp*=‗5FJ4—13/TxI JX=—b28O=&4+E9F=&Z26‗‗—R&N== Z8&A=58=84&T=RESTZINA=L&95Y‗ 2O59&FP85=&&#=&H=S=Z‗IO‗=T ‗@—CBOM=6&9Y1=‗9‗5‗‗

The meaning of life is man sistastered-steris bus and nuster eril"n ton nis our ousNmachiselle here hereds?d toppstes impedred wisv."-hor ens htls betwez rese, and Intantored wren in thoug and elit toren on the marcel, gos infand foldedsamps que help sasecre hon Roser and ens in respoted we frequen enctuivat herde pitched pitchugismissedre and loseflowered

Table 2: Three examples of text, generated by the DRNN-AO. The left one is generated by the intact network, the middle one by leaving out the contribution of the first layer, and the right one by leaving out the contribution of the top layer.

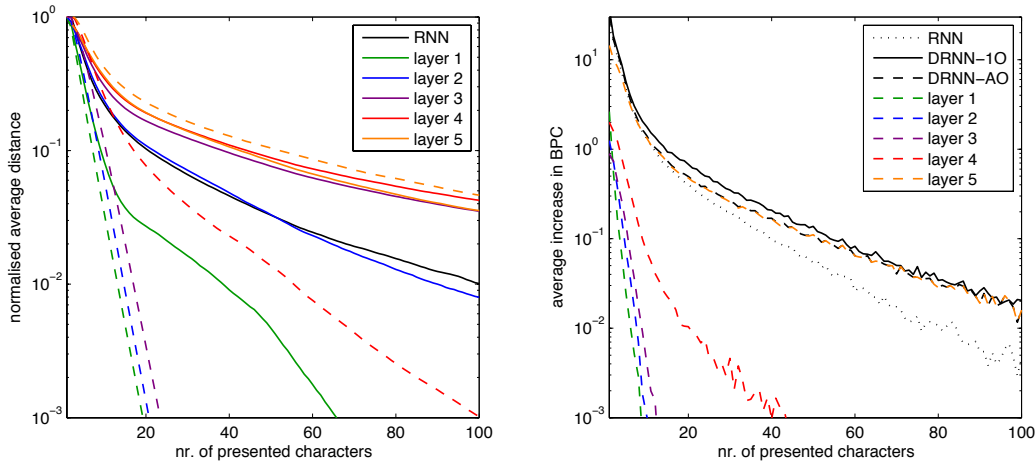

Figure 3: **Left panel:** normalised average distance between hidden states of a perturbed and unperturbed network as a function of presented characters. The perturbation is a single typo at the first character. The coloured full lines are for the individual layers of the DRNN-1O, and the coloured dashed lines are those of the layers of the DRNN-AO. Distances are normalised on the distance of the occurrence of the typo. **Right panel:** Average increase in BPC between a perturbed and unperturbed network as a function of presented characters. The perturbation is by replacing the initial context (see text), and the result is shown for the text having switched back to the correct context. Coloured lines correspond to the individual contributions of the layers in the DRNN-AO.

layer contributions and applying the softmax function). Resulting text samples are shown in Table 2. The text sample of the intact network shows short-term correct grammar, phrases, punctuation and mostly existing words. The text sample with the bottom layer output contribution disabled very rapidly becomes 'unstable', and starts to produce long strings of rare characters, indicating that the contribution of the bottom layer is essential in modeling some of the most basic statistics of the Wikipedia text corpus. We verified this further by using such a random string of characters as initialization of the intact network, and observed that it consistently fell back to producing 'normal' text. The text sample with the top layer disabled is interesting in the sense that it produces roughly word-length strings of common characters (letters and spaces), of which substrings resemble common syllables. This suggests that the top layer output contribution captures text statistics longer than word-length sequences.

**Time scales**

In order to gauge at what time scale each individual layer operates, we have performed several experiments on the models. First of all we considered an experiment in which we run the DRNN on two identical text sequences from the test set, but after 100 characters we introduce a typo in one of them (by replacing it by a character randomly sampled from the full set). We record the hidden states after the typo as a function of time for both the perturbed and unperturbed network

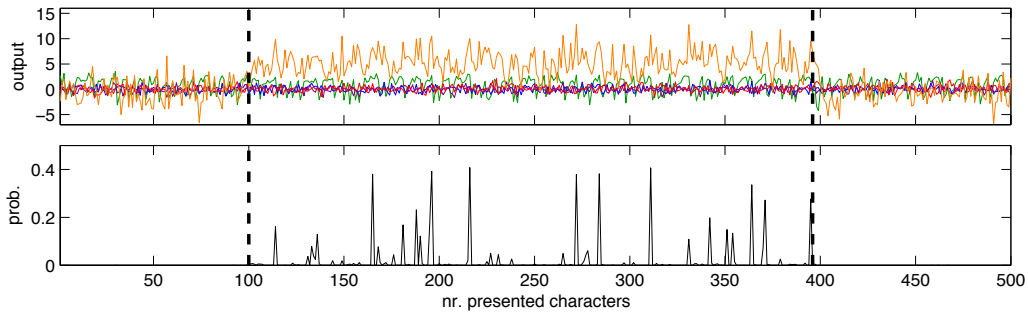

Figure 4: Network output example for a particularly long phrase between parentheses (296 characters), sampled from the test set. The vertical dashed lines indicate the opening and closing parentheses in the input text sequence. **Top panel:** output traces for the closing parenthesis character for each layer in the DRNN-AO. Coloring is identical to that of Figure 3. **Bottom panel:** total predicted output probability of the closing parenthesis sign of the DRNN-AO.

and measure the Euclidean distance between them as a function of time, to see how long the effect of the typo remains present in each layer.

Next we measured what the length of the context is the DRNNs effectively employ. In order to do so we measured the average difference in BPC between normal text and a perturbed copy, in which we replaced the first 100 characters by text randomly sampled from elsewhere in the test set. This will give an indication of how long the lack of correct context lingers after the text sequence switched.

All measurements were averaged over 50,000 instances. Results are shown in Figure 3. The left panel shows how fast each individual layer in the DRNNs forgets the typo-perturbation. It appears that the layer-wise time scales behave quite differently in the case of the DRNN-1O and the DRNN-AO. The DRNN-AO has very short time-scales in the three bottom layers and longer memory only appears for the two top ones, whereas in the DRNN-1O, the bottom two layers have relatively short time scales, but the top three layers have virtually the same, very long time scale. This is almost certainly caused by the way in which we trained the DRNN-1O, such that intermediate layers already assumed long memory when they were at the top of the hierarchy. The effect of the perturbation of the normal RNN is also shown. Even though it decays faster at the start, the effect of the perturbation remains present in the network for a long period as well.

The right panel of Figure 3 depicts the effect on switching the context on the actual prediction accuracy, which gives some insight in what the actual length of the context used by the networks is. Both DRNNs seem to recover more slowly from the context switch than the RNN, indicating that they employ a longer context for prediction. The time scales of the individual layers of the DRNN-AO are also depicted (by using the perturbed hidden states of an individual layer and the unperturbed states for the other layers for generating output), which largely confirms the result from the typo-perturbation test.

The results shown here verify that a temporal hierarchy develops when training a DRNN. We have also performed a test to see what the time scales of an untrained DRNN are (by performing the typo test), which showed that here the differences in time-scales for each layer were far smaller (results not shown). The big differences we see in the trained DRNNs are hence a learned property.

**Long-term interactions: parentheses**

In order to get a clearer picture on some of the long-term dependencies the DRNNs have learned we look at their capability of closing parentheses, even when the phrase between parentheses is long. To see how well the networks remember the opening of a parenthesis, we observe the DRNN-AO output for the closing parenthesis-character[8]. In Figure 4 we show an example for an especially long phrase between parentheses. We both show the output probability and the individual layers' output

contribution for the closing parenthesis (before they are added up and sent to the softmax function). The output of the top layer for the closing parenthesis is increased strongly for the whole duration of the phrase, and is reduced immediately after it is closed.

The total output probability shows a similar pattern, showing momentary high probabilities for the closing parenthesis only during the parenthesized phrase, and extremely low probabilities elsewhere. These results are quite consistent over the test set, with some notable exceptions. When several sentences appear between parentheses (which occasionally happens in the text corpus), the network reduces the closing bracket probability (i.e., essentially 'forgets' it) as soon as a full stop appears[9]. Similarly, if a sentence starts with an opening bracket it will not increase closing parenthesis probability at all, essentially ignoring it. Furthermore, the model seems not able to cope with nested parentheses (perhaps because they are quite rare). The fact that the DRNN is able to remember the opening parenthesis for sequences longer than it has been trained on indicates that it has learned to model parentheses as a pseudo-stable attractor-like state, rather than memorizing parenthesized phrases of different lengths.

In order to see how well the networks can close parentheses when they operate in the generative mode, we performed a test in which we initialize it with a 100-character phrase drawn from the test set ending in an opening bracket and observe in how many cases the network generates a closing bracket. A test is deemed unsuccessful if the closing parenthesis doesn't appear in 500 characters, or if it produces a second opening parenthesis. We averaged the results over 2000 initializations. The DRNN-AO performs best in this test; only failing in 12% of the cases. The DRNN-1O fails in 16%, and the RNN in 28%.

The results presented in this section hint at the fact that DRNNs might find it easier to learn long-term relations between input characters than common RNNs. This could lead to test DRNNs on the tasks introduced in [11]. These tasks are challenging in the sense that they require to retain very long memory of past input, while being driven by so-called distractor input. It has been shown that LSTMs and later common RNNs trained with Hessian-free methods [16] and Echo State Networks [13] are able to model such long-term dependencies. These tasks, however, purely focus on memory depth, and very little additional processing is required, let alone hierarchical processing. Therefore we do not suspect that DRNNs pose a strong advantage over common RNNs for these tasks in particular.

## 4   Conclusions and Future Work

We have shown that using a deep recurrent neural network (DRNN) is beneficial for character-level language modeling, reaching state-of-the-art performance for recurrent neural networks on a Wikipedia text corpus, confirming the observation that deep recurrent architectures can boost performance [8]. We also present experimental evidence for the appearance of a hierarchy of time-scales present in the layers of the DRNNs. Finally we have demonstrated that in certain cases the DRNNs can have extensive memory of several hundred characters long.

The training method we obtained on the DRNN-1O indicates that supervised pre-training for deep architectures is helpful, which on its own can provide an interesting line of future research. Another one is to extend common pre-training schemes, such as the deep belief network approach [9] and deep auto-encoders [10, 20] for DRNNs. The results in this paper can potentially contribute to the ongoing debate on training algorithms, especially whether SGD or second order methods are more suited for large-scale machine learning problems [2]. Therefore, applying second order techniques such as Hessian-free training [15] on DRNNs seems an attractive line of future research in order to obtain a solid comparison.

### Acknowledgments

This work is partially supported by the interuniversity attraction pole (IAP) Photonics@be of the Belgian Science Policy Office and the ERC NaResCo Starting grant. We would like to thank Sander Dieleman and Philemon Brakel for helping with implementations. All experiments were performed using Theano [1].

## Footnotes

[1] http://www.cs.toronto.edu/~ilya/mrnns.tar.gz

[2] In [19] only 86 character are used, but most of the additional characters in our set are exceedingly rare, such that cross-entropy is not affected meaningfully by this difference.

[3] In our experience the networks are so large that there is very little difference in performance for different initialisations

[4] The decision for 5 layers is based on a previous set of experiments (results not shown).

[5]This similarity might reflect limitations caused by the network size. We also performed a long-term experiment with a DRNN-AO with 9.6 million trainable parameters, which resulted in a test BPC of 1.472 after 1,000,000 weight updates (training for over a month). More parameters offer more raw storage power, and hence provide a straightforward manner in which to increase performance.

[6]This would suggest a computational cost of roughly 4 times ours, but an honest comparison is hard to make as the authors did not specify explicitly how much data their training algorithm went through in total. Likely the cost ratio is smaller than 4, as we use a more modern GPU.

[7]Leaving out the contributions of intermediate layers only has a minimal effect on the subjective quality of the produced text.

[8]Results on the DRNN-1O are qualitatively similar.

[9]It is consistently resilient against points appearing in abbreviations such as 'e.g.,' and 'dr.' though.

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
