[Reviews · NeurIPS 2013]

Submitted by Assigned_Reviewer_4

The authors propose a new deep architecture, which combines the hierarchy of deep learning with time-series modelling known from HMMs or recurrent neural networks. The proposed training algorithm builds the network layer-by-layer using supervised (pre-)training a next-letter prediction objective. The experiments demonstrate that after training very large networks for about 10 days, the network performance on a Wikipedia dataset published by Hinton et al. improves over previous work. The authors then proceed to analyze and discuss details of how the network approaches its task. For example, long-term dependencies are modelled in higher layers, correspondence between opening and closing parenthesis are modelled as a “pseudo-stable attractor-like state”.

The paper is of good quality. It is well-written, experiments are set up well and the figures support the findings well. My main issue with the paper is the experiments. The comparison is done on one dataset only, which does not seem to have much work on it. Direct comparison is only with other RNN-related models, not with maybe more common hierarchical HMMs etc., which are able to solve similar tasks (Tab. 1 lists other approaches, but on a different corpus, which makes them incomparable).

The contribution of the paper is more in the analysis of /how/ the network operates (which is analyzed quite well) than in what it achieves, but this is not what the paper sets out for.
Summary: A new deep architecture is proposed, combining hierarchical features of deep learning with time-series modelling. The paper is well-written, the model is analyzed well, but results are somewhat inconclusive.

Submitted by Assigned_Reviewer_6

REPLY TO REBUTTAL:
- I mostly agree with the one task comment.
- It would still be worth trying truncated backprop with e.g., 30 steps. It is less likely to learn to balance parentheses (but it would be good to verify that), but it may still achieve competitive entropies at a fraction of the cost.


The usefulness of deep recurrent neural networks is investigated.
The paper reaches interesting conclusions: when holding the number
of parameter constant, deeper recurrent neural networks outperform
standard RNNs. Other interesting findings include an analysis
that shows the deeper networks to contain more "long term information",
and the result that plain BPTT can train character-level RNNs
to balance parentheses, where previously it was thought to be possible
only with HF.

But it claims to introduce the deep RNN, but those been used before;
for example, in http://www.cs.toronto.edu/~graves/icassp_2013.pdf, and likely
much earlier than that. So the paper should not claim to have introduced this
architecture. It is OK to not introduce new architectures and to focus
on an analysis.

The analysis, while meaningful, would be even more interesting if it
were shown on several problems, perhaps something on speech. Otherwise
the findings may be an artifact of the problem.

Finally, the HF experiments were likely too expensive, since the
truncated backprop approach (that was introduced in the 90) was
successfully used by Mikolov (see his PhD thesis) to train RNNs
to be excellent language models. Thus it is likely that truncated
BPTT would do good here too, and it would be nice to obtain a
confirmation of this fact.
Summary: This work presents an interesting analysis of deep RNNs. The two main results are: 1) deep RNNs outperform standard RNNs when the number of parameters is fixed on character-level language modelling; 2) the deeper layers exhibit more long-range structure; and 3) truncated BPTT can train character-level RNNs to balance parentheses.

The results are interesting, but given that it is only an analysis, the paper would be a lot stronger if it had a similar analysis on some speech task (for example, to show that deeper RNNs do better, and to show that their deeper layers have more long range information)>

Submitted by Assigned_Reviewer_7

I like the point about the arbitrariness of older data benefiting from potentially many more layer of processing compared to newer data. And building a deep RNN seems like a reasonable way to do better on shorter term stuff, in addition to possible other benefits. Another obvious thing to do would be to put extra layers of nonlinearities between each timestep. Have you tried this?

How important was the gradient normalizations and why does this "avoid of the problem of bifurcations"? There is a recent paper in ICML 2013 by researchers from U of Montreal that looks at gradient truncation and optimization in RNNs that that may be relevant here. Also, will normalizing the gradient in this way potentially mess up the "averaging" behavior of SGD?

In terms of previous results on these kinds of Wikipedia compression tasks, there is also some work by Mikolov et al. that you may want to compare to.

Of the various experiments designed to examine the different roles played by each layer in terms of time-scale and perhaps "abstraction", the one I find most persuasive is the text generation one (as shown in Table 2). However, as pointed out by the authors themselves in the paragraph on line 241, it may be problematic to interpret the effect of this kind of "brain surgery" on RNNs due to the complex interdependencies that may have developed between the outputs of the various layers.


For me, the biggest missing piece of the empirical puzzle in this paper is the question of whether the higher layers are actually *better* at processing more abstract and long-term properties of the text than if the units were moved to the first layer. i.e. are they benefiting from the extra levels of processing that proceed them in a nontrivial way? That they happen to take on these seemingly more abstract and longer term roles after training is good but incomplete evidence that this is the case.

I notice that the regular RNN seems to have less parameters in these experiments since 2*767^2 * 5 > 2119^2 (I'm counting both recurrent and inter-layer weight matrices, hence the multiplication by 2), so the comparison might be a bit unfair. A more convincing would be if the deeper RNNs did better than a standard RNN with the same number of parameters, or even better, the same number of *units*. Say a 2 layer DRNN versus such an RNN with the same number of units, where the different in the number of parameters wouldn't favor the RNN so much that the comparison would be rendered unfair. This would be strengthen the paper's claims a lot in my opinion.

Also, instead of Figure 2, it would be better to have seen how well various depths of DRNN did on the benchmarks when trained from scratch, possibly with wider layers than their deeper counterparts to make up for the difference the numbers of parameters and/or units.



Minor:
- You should define DRNN-AO and IO in the text somewhere and not just in Figure 1.
Summary: This paper looks at a hybrid of deep and recurrent neural networks called DRNNs, which are like deep networks but with recurrent connections at each layer that operate through time. The authors show how such an architecture can work very well for text prediction/compression compared to existing approaches.

A large bulk of this paper is devoted to a series of experiments designed argue that the higher level layers are processing/representing more abstract and long-term structures in the data. These experiments are pretty convincing but I have a few reservations, as elaborated on below in my full review, and would like to see a couple more experiments.

I think that it is worthwhile to look at these kinds of deep temporal networks and to gain insight into how they function after training. The paper is also easy to read, and seems quite intellectually honest and thorough about its own potential problems and shortcomings, which is something I especially appreciate.
Author Feedback

Author rebuttal: GENERAL REMARKS:

ONE TASK ONLY: Concerning the remarks on using only one dataset: the main reason for this is the page limit, and the fact that this is a difficult real-world large-scale dataset in which we know a temporal hierarchy definitely exists. We chose to verify our claims on this task properly, rather than spending the available space on presenting more superficial findings on several tasks. In the past we have used (relatively small) DRNNs on the TIMIT speech corpus, which confirm the advantage of multiple layers (though we didn't reach state-of-the-art). In fact, the paper by Graves mentioned by reviewer 6, is on TIMIT, and they reach extremely good performance by stacking LSTMs, in virtually the same way as we do with RNNs. Even though this result is obfuscated by the fact that LSTMs are somewhat different from RNNs, deep architectures greatly improve performance here too, which strengthens our claim. I suggest that we mention this point in the discussion.


MODEL SELECTION: There seems to be some confusion on this issue: the number of nodes per layer for each model was carefully selected such that all models had a total of (as close as possible to) 4.9 million trainable parameters, this in order to make a comparison to other work in literature.
The DRNNs are defined by 5 recurrent weight matrices, 4 interlayer weight matrices, a variable set of output weights (1 or 5 for the DRNN1O and -AO respectively) and a set of input weights (this is Z_1). Including bias terms we obtain:
DRNN-1O: input weights: 727*97; interlayer: 4*727*728; recurrent: 5*727^2; output: 728*96; total: 4900072
DRNN-AO: input weights: 706*97; interlayer: 4*706*707; recurrent: 5*706^2; output: 5*707*96; total: 4896590
RNN: input weights: 2119*97; recurrent: 2119^2; output: 2120*97; total: 4899224
Concerning model selection; the choice of 5 layers was based on earlier experiments (performed in a somewhat different way, such that they don’t fit in the paper as it is now), but 4 or 6 layers would probably give similar results.
Both these points can be made clearer by changing the text in minor ways.


REVIEWER 4

The meta-parameters we optimised (crudely, given the length of a single experiment) were those defining the training algorithm (initial learning rate vs. number of training examples) and initial scaling for weight matrices. Luckily, due to the fact that this is such a large-scale problem, there is little difference between test and train error, and we can safely rely on training errors for meta-parameter optimization.


REVIEWER 6

- Indeed, the concept of DRNNs is not entirely novel (though we do in fact refer to the paper suggested by the reviewer). We will change the text in order to make this clearer.

- Note that we essentially have used truncated BPTT (T-BPTT), (by running many "short" (250 character) sequences in parallel). Commonly, T-BPTT is concerned with even shorter sequences (order of 10), but for character-level modeling this would mean the training can only take into account word-length sequences, and we would not be able to obtain the long-term dependencies as shown in the paper. The reference given by the reviewer is on word-based language modeling, where short sequences contain far more meaning.


REVIEWER 7

- Adding extra nonlinear layers between time steps is indeed an idea we considered, but in the end didn’t implement. For one it would be slow in execution (as there is less low-level parallelism to take advantage of), and such an architecture would likely suffer much more from fading gradient problems, as it would entail a network which is N times as deep as a common RNN folded out in time, N being the number of layers between each time steps.

- RNNs can exhibit extremely large gradients very suddenly (associated with being at or near a bifurcation). Simply using this gradient would lead to a very large jump in parameter space, leading to unpredictable and usually catastrophic results. We have also tried truncation, but this seemed to impede performance in the end. The bifurcations themselves are not the main problem, but the size of the gradient is. I suggest rewriting the sentence in question to make this clearer.
Indeed, the average of the normalized gradient is not the same as the normalized average gradient, such that there will be some difference between the two, but I suppose the same point can be made about truncated gradients.

- Indeed, the experiments are so far only suggestive, and not full affirmation of increasing abstraction. We are currently contemplating better experiments to gauge the level of abstraction of each layer. One potential way is to sample optimal text sequences for individual node activations at different layers, and see whether these show higher levels of abstraction higher in the DRNN, but this is non-trivial due to the discrete nature of the input. Due to page restrictions (and time limitations) we will not be able to include this in the paper.

- The DRNN-1O / AO is defined in the text at the start of section 2.2.